# Monitoring and Exposure Assessment of Fosetyl Aluminium and Other Highly Polar Pesticide Residues in Sweet Cherry

**DOI:** 10.3390/molecules28010252

**Published:** 2022-12-28

**Authors:** Emrah Gormez, Ozgur Golge, Miguel Ángel González-Curbelo, Bulent Kabak

**Affiliations:** 1Pia Frucht Food Control Laboratory, Alaşehir 45600, Turkey; 2Department of Gastronomy and Culinary Arts, Faculty of Tourism, Alanya Alaaddin Keykubat University, Alanya 07450, Turkey; 3Departamento de Ciencias Básicas, Facultad de Ingeniería, Universidad EAN, Calle 79 nº 11-45, Bogotá 110221, Colombia; 4Department of Food Engineering, Faculty of Engineering, Hitit University, Corum 19030, Turkey

**Keywords:** analytical method validation, chromatography, food safety, mass spectrometry, polar pesticides, risk assessment

## Abstract

Cherries are popular fruits due to their health benefits, organoleptic quality, and attractive appearance. Since highly polar pesticides are of low mass and amphoteric character, and are not amenable to traditional multi-residue extraction methods, they are more commonly not included in the pesticide monitoring program. This study aims to determine twelve highly polar pesticide residues in cherry samples intended for export from Turkey. A total of 16,022 cherry samples from 2018–2020 harvests in four production areas of Turkey were analyzed using a modification of the Quick Polar Pesticides method and liquid chromatography-tandem mass spectrometry. The method was validated at two fortification levels (0.01 and 0.05 mg kg^−1^), and good recoveries (87.4–111.4%) and relative standard deviations (<6%) were achieved for all analytes. The limits of quantification were in the range of 1.08–2.55 μg kg^−1^. Overall, 28.4% of the analyzed cherry samples were detected with phosphonic acid, calculated as fosetyl aluminium (fosetyl-Al) in amounts up to 77.7 mg kg^−1^. For 2304 samples (14.4%), the residues exceeded the European Union maximum residue level of 2 mg kg^−1^. There is no reason to be concerned about long-term exposure to phosphonic acid/fosetyl-Al, and the other highly polar pesticides through the consumption of sweet cherry.

## 1. Introduction

Cherry is one of the most consumed fruit throughout the world, which belongs to the genus *Prunus*, under the *Rosaceae* family. While it has been identified more than thirty cherry spices, mainly native to Europe and West Asia, the sweet cherry (*Prunus avium*) and sour cherry (*Prunus cerasus*) are globally traded. Cherries are good sources of fibre, potassium, polyphenolics (anthocyanins and hydroxycinnamic acid), β-caratone, and vitamin C and possess a high antioxidant capacity. Sweet cherries are mostly consumed as fresh fruit, whereas sour cherries are most frequently incorporated in processed foods, such as juices, jam, jellies, pies, cakes, ice cream, and others [1,2]. Turkey is the world’s leading producer of sweet cherries, producing 639,564 metric tonnes in 2018 and accounting for over 25% of total world production, followed by the United States (312,430 tonnes), Uzbekistan (172,035 tonnes), Chile (155,935 tonnes), and Iran (137,268 tonnes) [3]. Turkey exported cherries to the value of more than 150 million dollars in 2018 and accounting for 15% of global exports, the main importers being the Russian Federation and Germany [4].

Cherry is affected by insect and mite pests and by fungal, bacterial, and viral diseases during its growing process. It is susceptible to many diseases, including brown rot of stone fruit (*Monilinia fructigena*), brown rot of blossom (*Monilinia laxa*), *Armillaria* root rot (*Armillaria mellea*), *Phytophthora* root and crown rot (*Phytophthora* spp.), cherry leaf spot (*Blumeriella jaapi*), cherry shot-hole (*Stigmina carpophila*), bacterial canker (*Pseudomonas syringae*), and crown gall (*Agrobacterium tumefaciens*) [5].

The most effective way to prevent, destroy or control harmful organisms and diseases in the crop is by using pesticides. Within the agricultural sector in Turkey, most pesticides utilized in 2018 were from fungicides/bactericides (42.5%), followed by insecticides (29.6%) and herbicides (27.3%) [3]. However, the residues in agricultural products are a growing concern because of their adverse acute and chronic health effects and environmental problems. The fungicides azadirachtin, cyprodinil, dithiocarbamates, fludioxonil, tebuconazole, and thiophanate, the insecticides deltamethrin, dimethoate, pirimicarb, 1pinosad, spirodiclofen, tau-fluvalinate and thiacloprid, and highly polar herbicides are the most used pesticides in the cultivation of cherries in Turkey [6].

The use of highly polar pesticides In agriculture and horticulture is widespread due to their low costs, high efficiency, low persistence in the environment, and relatively low toxicity in comparison with other pesticides towards mammals [7]. However, polar pesticides are more commonly not included in national pesticide monitoring programs as they have low mass, amphoteric character, and are not amenable to traditional multi-residue methods. Conventionally, a series of single residue methods are used to detect and quantify, which resulted in extra costs, time delays, and excluded from the surveillance program. For this reason, a fast and simple single analytical method that can analyze multi-residue polar pesticides in agricultural products and detect maximum residue level (MRL) violations with confidence is in great demand.

The Quick Polar Pesticides (QuPPe) method established by the European Reference Laboratory-Single Residue Methods (EURL-SRM) allows the simultaneous extraction of highly polar pesticides from a wide range of food commodities. This method involves extraction with acidified methanol without clean-up and liquid chromatography-tandem mass spectrometry (LC-MS/MS) measurement [8]. Several studies have been conducted in various countries to monitor multiple highly polar residues in vegetables and fruits [7,9,10,11], cereals [12], animal-derived products [13], honey [11], low alcoholic beverages [14] and human blood serum [15] in the last five years.

In this study, we aimed to determine twelve highly polar residues, namely aminomethylphosphonic acid (AMPA), N-acetyl-AMPA, chlorate, ethephon, ethephon-hydroxy (HEPA), fosetyl-aluminium (fosetyl-Al), glyphosate, glufosinate, N-acetyl-glufosinate, maleic hydrazide, 3-methylphosphinicopropionic acid (MPPA) and phosphonic acid in Turkish cherries intended for export to various countries mainly to Russian Federation and European countries. For that purpose, a modified method based upon the QuPPe extraction method followed by an LC-MS/MS measurement was validated.

## 2. Results and Discussion

### 2.1. Validation Data

The validation results for the cherry matrix compared favorably against the analytical performance described in the SANTE 11813/2017 guideline. As shown in Table 1, the coefficients of determination and the residuals were excellent (R^2^ > 0.99 and residuals <20%). The limits of quantification (LOQs) ranged from 1.08 μg kg^−1^ for phosphonic acid to 2.55 μg kg^−1^ for glyphosate. The LOQs of target polar compounds were much lower than the European Union (EU) MRLs in cherry. According to Table 2, all recoveries were satisfactory, with mean values ranging from 87.4% to 111.4%, and relative standard deviations (RSD) values varying from 0.47 to 5.12% under repeatability conditions and from 1.68 to 5.04% under reproducibility conditions for target polar compounds, demonstrating good repeatability of the measurements in the absence/presence of ILIS. For all polar compounds, U_exp_ was greatly lower than the criteria of 50% specified in SANTE 11813/2017 guideline. The U_exp_ ranged between 7% for chlorate and fosetyl-Al and 27% for glyphosate.

### 2.2. Pesticide Analysis in Sweet Cherry Samples and Exposure Assessment

In total, 16,022 cherry samples from 2018, 2019 and 2020 harvests were monitored for the twelve highly polar pesticides. All the cherry samples were produced in four cherry production areas in Turkey. None of the target polar compounds was measured above the LOQs in cherries except for phosphonic acid, calculated as fosetyl-Al, sum. The frequency of cherry samples with fosetyl-Al residue produced in different areas of Turkey is shown in Figure 1. In the harvest years 2018, 2019, and 2020, 6.3–24.2%, 13.9–38.2%, and 17.5–43.4% of the samples, respectively, contained fosetyl-Al at different concentrations. In all three sampling years, Izmir samples had the most frequency of fosetyl-Al residues (24.2–43.4% frequency) in cherries, followed by Isparta samples (21.2–41.4%).

Figure 2 reveals the distribution of fosetyl-Al in sweet cherry samples, taking into consideration the harvest years. Overall, 78.3% of cherry samples harvested in 2018 were free from fosetyl-Al residue, while 12.2% of samples contained fosetyl-Al at levels not exceeding the respective EU MRL of 2 mg kg^−1^. The level of fosetyl-Al exceeded the legal limit in 410 cherry samples (9.5%). No fosetyl-Al was found in 75.6% of cherry samples from the 2019 harvest, whereas 9.8% of the samples tested contained quantified residue of fosetyl-Al not exceeding the respective EU MRL. In 808 cherry samples (14.6%), fosetyl-Al residue levels exceeded the EU MRL. Cherry samples from the harvest of the year 2020 had a high frequency of fosetyl-Al (36.8%) compared to the other years. Out of 3354 quantified samples of cherries from the 2020 harvest, 1086 samples (17.6%) showed fosetyl-Al concentrations above the EU MRL. The concentrations of fosetyl-Al in cherries collected in three consecutive years, 2018–2020, varied from 0.013 to 18.6 mg kg^−1^ (mean = 0.862 mg kg^−1^), from 0.005 to 16.9 mg kg^−1^ (mean = 0.308 mg kg^−1^) and from 0.005 to 77.7 mg kg^−1^ (mean = 0.432 mg kg^−1^), respectively. LC-MS/MS chromatograms of the extract of the cherry sample containing phosphonic acid (calculated as fosetyl-Al) at a level of 1.73 mg kg^−1^ are illustrated in Figure 3.

The use of fosetyl-Al is not common among cherry farmers in Turkey. Farmers declared that they do not use fosetyl-Al contrary to foliar fertilizer. Residues of phosphonic acid, defined as fosetyl-Al, in cherries could occur as a result of applying foliar fertilizer containing phosphonic acid itself prior to harvest.

These results are inconsistent with the 2015 EU pesticide monitoring program findings performed by the EU Member States, Iceland, and Norway. Fosetyl-Al was present in 29.9% of 84,341 samples in quantifiable concentrations; 1.21% of them (59 samples) exceeded the EU MRL. Cherries were also found to contain different pesticides in 177 out of 719 samples analyzed (24.6%), 3.2% of which exceeded the respective MRLs [16]. In another extensive study, a total of 785 fresh fruit samples (including 23 sweet cherry samples) from conventional cultivation were analyzed by CVUA Stuttgart for over 750 different pesticides. Fosetyl, a sum fungicide was found to be the predominant pesticide detected in fresh fruits (47.4% of the samples analyzed) from 40 different countries, up to a level of 47.7 mg kg^−1^. Cherries had fosetyl, sum, at concentrations varying from 0.083 to 2.1 mg kg^−1^ [17]. In contrast to our results, fosetyl-Al was not determined in any 225 sweet cherry samples consumed domestically in Turkey [18]. In a study by Da Silva et al. [7], ethephon was found in 547 out of 1048 fruits (53%) intended for export from Brazil; 17 of them (2%) had residues higher than the legal limit. Fosetyl was also detected in 20 out of 109 mango samples (18%) in measurable concentrations. The level of fosetyl exceeded the respective MRL in 4.6% of the mango samples. During the years 2004–2011, the Danish Veterinary and Food Administration monitored 17,309 food commodities, including fruits, vegetables, cereals, and animal origin products, for about 250 pesticides, but polar pesticides were not included in the monitoring program. Cherry samples (*n* = 24) were found to contain various pesticides, including bifenthrin (4.2% of the cherries), carbendazim (20.8%), lambda-cyhalothrin (8.3%), cypermethrin (16.7%), cyprodinil (4.2%), diazinon (8.3%), dimethoate (8.3%), iprodione (4.2%), monocrotophos (4.2%), myclobutanil (16.7%), and tebuconazole (12.5%) [19].

Fosetyl-Al is a systematic fungicide that has been used to protect many fruits and vegetables against plant pathogens such as *Phytophthora*, *Pythium*, *Plasmopara*, *Bremia* spp. as well as bacteria such as *Xanthomonas* and *Erwinia* spp. [20]. Fosetyl-Al does not show carcinogenic, genotoxic, or mutagenic properties in laboratory animals, and it does not pose developmental or reproductive effects of concern. An acceptable daily intake (ADI) of 3 mg kg^−1^ body weight (b.w.) per day and an acceptable operator exposure level (AOEL) of 5 mg kg^−1^ b.w. per day for fosetyl-Al has been established. The ADI of 2.52 mg kg^−1^ b.w. has also been set for phosphonic acid, expressed as fosetyl [21].

The mean long-term exposure to phosphonic acid/fosetyl from sweet cherry for adults ranged from 1.55 × 10^−5^ to 1.56 × 10^−5^ mg kg^−1^ b.w. day^−1^ (LB to UB). This is the first data on long-term exposure to phosphonic acid/fosetyl through the consumption of sweet cherries for adults. Since all sweet cherry data were left-censored for other highly polar substances analyzed, those exposure estimates were not included in the analysis. The HQ of fosetyl for adults was 0.0006% (LB/MB/UB). Applying the long-term exposure assessment method, none of the samples exceeded the toxicological reference value (max. 0.34% of the ADI) for fosetyl. There is, therefore, no reason to be concerned about long-term exposure to residues, phosphonic acid/fosetyl, and other highly polar substances through the consumption of sweet cherries.

## 3. Materials and Methods

### 3.1. Chemicals and Materials

LC-MS grade acetonitrile and methanol were supplied by J.T. Baker (Gliwice, Poland) and VWR Chemicals BDH^®^ (Gdansk, Poland), respectively. Formic acid and glacial acetic acid were ordered from Merck KGaA (Darmstadt, Germany).

The analytical standards of AMPA (purity of 99.9%), ethephon (96.0%), HEPA (89.5%), fosetyl-Al (95.0%), glyphosate (98.7%), glufosinate (97.9%), N-acetyl-glufosinate (94.3%), maleic hydrazide (99.0%), MPPA (99%) and phosphonic acid (97.5%) were obtained from Dr. Ehrenstorfer GmbH (Augsburg, Germany). N-acetyl-AMPA (94.4%) and chlorate (99.0%) were from HPC Standards GmbH (Cunnersdorf, Germany). Isotopically labelled internal standards (ILISs) ethephon D4 (94.3%) and fosetyl-Al D15 (96.4%) were supplied from Dr. Ehrenstorfer GmbH (Augsburg, Germany). The ILISs of glyphosate-^13^C_2_,^15^N (>95%) and ^18^O_3_-phosphonic (≥95%) were obtained from EURL-SRM (Stuttgart, Germany) and Toronto Research Chemicals (Toronto, ON, Canada), respectively.

### 3.2. Samples

A total of 16,022 cherry samples, each weighing 2 kg harvested for export, were collected from Turkey for the analysis of the twelve highly polar residues. The samples were originating from four Turkish cherry production areas, namely İzmir-Kemalpaşa, Denizli, Isparta, and Afyon. Sampling was carried out for three consecutive years, 2018–2020, and yearly size varied between 4319 and 6170 samples. Each analytical result was derived from one laboratory sample taken from each lot.

### 3.3. Sample Preparation

Sweet cherry samples were extracted using the EURL-SRM QuPPe method [8], with slight modifications. The extraction procedures were schematically depicted in Figure 4. Briefly, ten grams of homogenized cherry samples were placed into 50 mL polypropylene centrifuge tubes, and 1.5 g of water was added and spiked with 50 μL of ILIS solution. Then, 10 mL of acidified MeOH (containing 1% formic acid, *v*/*v*) were added, shaken for 2 min in a Collomix shaker (VIBA 330, Gaimersheim, Germany), and the tubes were centrifuged (Rotofix 32 A, Hettich, Tuttlingen, Germany) for 5 min at 4000 rpm at room temperature. Formic acid was used for the adjustment of pH. Finally, 1 mL of supernatant was filtered through a regenerated cellulose syringe filter (0.20 μm) and collected in plastic autosampler vials.

### 3.4. LC-MS/MS Analysis

The LC-MS/MS system comprised of an Agilent 1290 LC coupled to an Agilent 6470 triple quadrupole (QQQ) mass spectrometer (Agilent, Santa Clara, CA, USA) equipped with a Jet Stream electrospray ionization (ESI) source. Instrument control, data acquisition, and quantitative analysis were performed using the Agilent MassHunter workstation software. Separation of highly polar compounds was achieved using a porous graphitic carbon-based Thermo Scientific™ Hypercarb column (100 × 2.1 mm, 5 μm particle size) at 40 °C. Eluent A composed of water containing 5% methanol and 1% acetic acid, and eluent B is composed of methanol containing 1% acetic acid. Gradient elution was performed as follows: 0–11 min 100–70% A, 0.2 mL min^−1^; 11–19 min 70% A, 0.4 mL min^−1^; 19–22 min 10% A, 0.4 mL min^−1^; 22.1–30 min 100% A, 0.2 mL min^−1^. The injection volume was 10 μL.

Electrospray negative ionization (ESI-) was used for the monitoring of the twelve highly polar compounds and the four ILISs. The multiple reaction monitoring (MRM) settings for each polar compound were optimized by infusing neat standard solutions. The parameters for each target analyte’s MRM transition are given in Table 3.

### 3.5. Validation Studies

The performance of the modified QuPPe method was assessed using SANTE/11813/2017 guideline [22]. Matrix-matched multi-residue calibration standards were constructed by adding six different concentrations (5, 10, 25, 50, 50, 100, and 250 μg kg^−1^) of each polar compound in the blank cherry extract. The calibration curves for each target analyte were prepared by running matrix-matched calibration standards, and R^2^ values of >0.99 were acceptable. The method’s precision (repeatability and within-laboratory reproducibility) and accuracy were assessed by analyzing blank cherry samples fortified with 0.01 and 0.05 mg kg^−1^ for analytes. The analysis was performed in five replicates (*n* = 5) at each level. The LOQs were determined as the lowest concentration that provided an accuracy rate of 70–120% and RSD of ≤20%. Two sources of uncertainty (uncertainty associated with trueness (bias) and within-laboratory reproducibility) were considered in the determination of expanded measurement uncertainty (U_exp_) for each analyte, as described in detail previously [23].

### 3.6. Exposure Analysis and Risk Assessment

The long-term dietary exposure to highly polar substances from the consumption of sweet cherries was calculated by multiplying the residue concentration by sweet cherry consumption data (Equation (1)) [24].
(1)$$Dietary\;exposure\;=\;\frac{Concentration\;of\;residue\;infood\;\left(\frac{mg}{kg}\right)x\;Food\;consumption\;\left(kg/day\right)\;}{Body\;weight\;\left(kg\right)}$$

The non-detect results were treated by the substitution method as described in the EFSA Scientific Report [25]. The left-censored results were input as “zero”, “a value of the respective LOQ”, and “LOQ/2” according to Lower Bound (LB), Upper Bound (UB), and Middle Bound (MB) scenarios, respectively.

The consumption rate of sweet cherry (0.1107 g kg^−1^ b.w. day^−1^) from the GEMS/Food G06 cluster diets and a standard body weight of 60 kg have been assumed to calculate dietary exposure to highly polar residues for adults [26].

To assess the health risks of polar residues, the Hazard Quotient (HQ), which was calculated by dividing the potential exposure to a chemical hazard by the reference dose (Equation (2)) as described by Reffstrup et al. [27].
(2)$$Hazard\;Quotient\;\left(HQ\right)=\frac{Exposure\;of\;theconcerned\;residue}{Reference\;value\;\left(ADI\right)}$$

## 4. Conclusions

This study was conducted to monitor twelve highly polar pesticides in sweet cherries intended for export from Turkey to various countries, mainly Russia and European countries. A modified QuPPe method was successfully validated and applied for the analysis of 16,022 cherry samples from 2018–2020 harvests. Among the polar compounds, only phosphonic acid residues, calculated as fosetyl-Al, sum, were detected in cherry samples. Fosetyl-Al was measured in 28.4% of the cherry samples in quantifiable concentrations; 2304 of these samples (14.4%) had fosetyl-Al above the MRL. There is no health risk in the consumption of sweet cherries intended for export from Turkey.

## Figures and Tables

**Figure 1 molecules-28-00252-f001:**
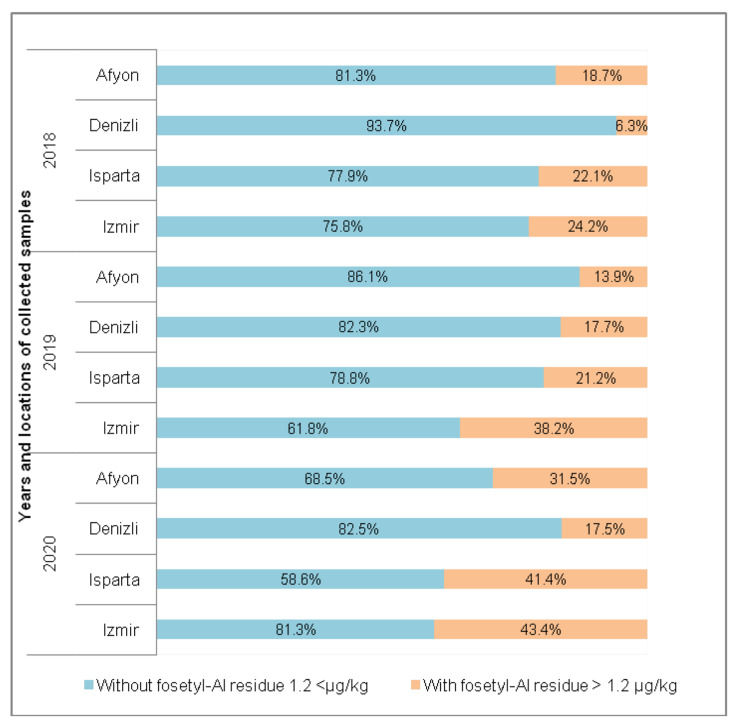
The frequency of cherry samples with fosetyl-Al per production area and harvest year.

**Figure 2 molecules-28-00252-f002:**
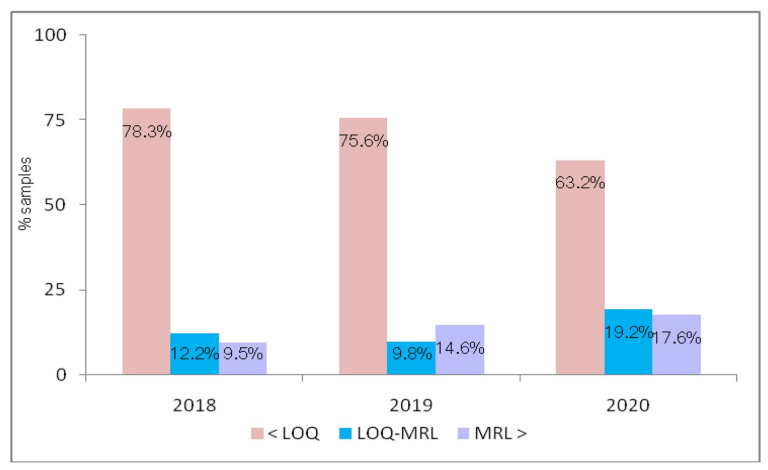
The distribution of fosetyl-Al content in cherry samples per harvest year.

**Figure 3 molecules-28-00252-f003:**
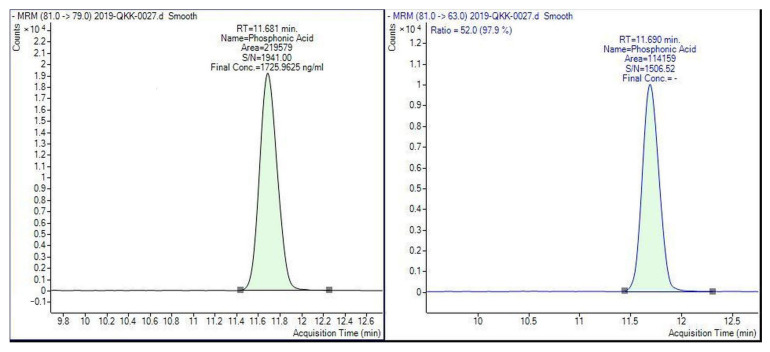
LC-MS/MS chromatograms of the extract of the cherry sample containing phosphonic acid (calculated as fosetyl-Al) at a level of 1.73 mg kg^−1.^

**Figure 4 molecules-28-00252-f004:**
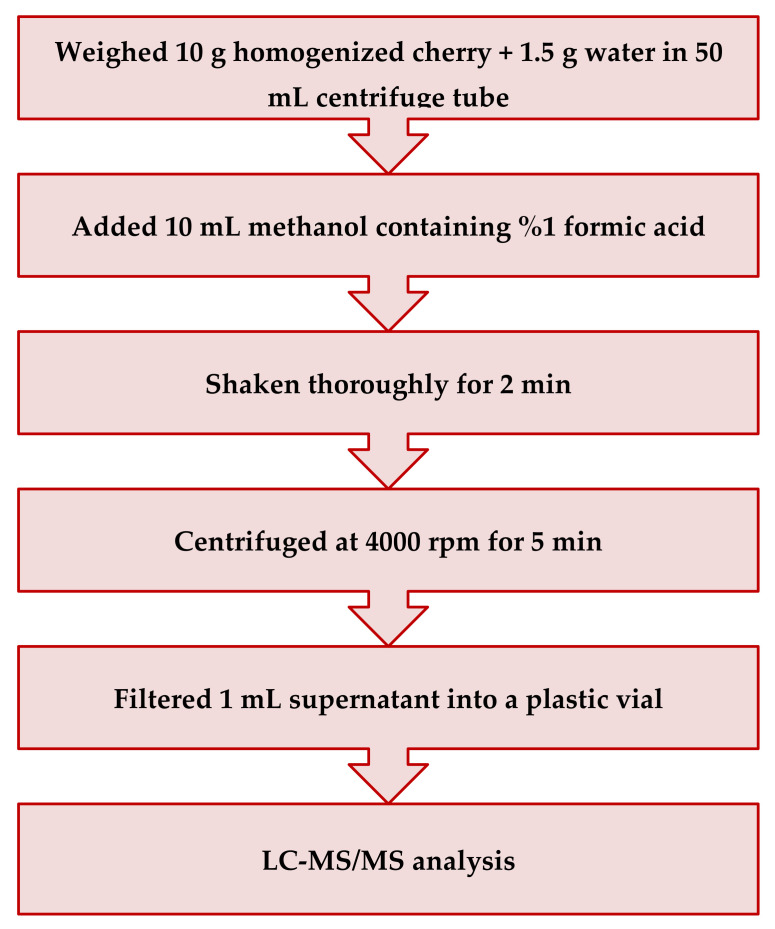
The schematic diagram of the QuPPe method.

**Table 1 molecules-28-00252-t001:** LOQs, EU MRLs and matrix-matched calibration data of the highly polar pesticides.

Analyte	LOQ (μg kg^−1^)	EU MRL (mg kg^−1^)	Linearity (Range: 5–250 μg kg^−1^)		Residual (%)
			Equation	*R* ^2^	
AMPA	1.76	None	*y* = 4618*x* − 4087	0.999	<12
N-acetyl-AMPA	2.19	None	*y* = 21,781*x* − 19,397	0.994	<16
Chlorate	1.46	0.05	*y* = 30,155*x* − 14	0.994	<16
Ethephon	1.10	5.0	*y* = 43,871*x* − 54,173	0.996	<14
HEPA	1.26	None	*y* = 39,407*x* − 102,623	0.999	<12
Fosetyl-Al	1.18	2.0 ^b^	*y* = 77,048*x* − 103,586	0.999	<13
Glyphosate	2.55	0.1	*y* = 9506*x* − 11,750	0.998	<11
Glufosinate	2.23	0.15 ^a^	*y* = 3071*x* − 5666	0.995	<20
N-acetyl-glufosinate	1.21	Part of glufosinate	*y* = 13,979*x* − 29,314	0.999	<12
Maleic hydrazide	2.14	0.2	*y* = 2430*x* + 1373	0.998	<18
MPPA	1.40	Part of glufosinate	*y* = 26,002*x* − 26,203	0.995	<13
Phosphonic acid	1.08	Part of fosetyl-Al	*y* = 28,624*x* − 68,551	0.998	<14

^a^ Sum of glufosinate, its salts, MPPA and N-acetyl-glufosinate expressed as glufosinate equivalents. ^b^ Sum of fosetyl, phosphonic acid and their salts expressed as fosetyl.

**Table 2 molecules-28-00252-t002:** Recovery, precision and expanded uncertainty of the highly polar pesticides.

Analyte	Recovery (%)		Repeatability (%RSD, *n* = 5)		Reproducibility (%RSD, *n* = 10)		*U*_exp_ (%)
	0.01 (mg kg^−1^)	0.05 (mg kg^−1^)	0.01 (mg kg^−1^)	0.05 (mg kg^−1^)	0.01 (mg kg^−1^)	0.05 (mg kg^−1^)	
AMPA	98.0	95.3	3.94	2.98	5.04	3.74	16
N-acetyl-AMPA	92.2	95.7	3.36	2.38	2.47	1.77	17
Chlorate	101.1	102.9	1.53	1.35	2.31	1.68	7
Ethephon	94.7	96.0	3.58	2.21	1.97	2.30	14
HEPA	106.3	94.5	2.90	1.12	3.11	2.36	12
Fosetyl-Al	102.4	97.9	0.94	0.47	2.47	1.83	7
Glyphosate	89.4	95.7	5.12	2.00	3.90	3.42	27
Glufosinate	99.8	87.4	4.19	2.39	2.25	2.17	19
N-acetyl-glufosinate	100.6	92.3	4.08	2.05	2.77	2.13	16
Maleic hydrazide	103.2	111.4	3.25	2.37	3.55	2.49	20
MPPA	95.0	99.5	1.81	2.09	3.47	2.19	11
Phosphonic acid	96.8	90.6	2.69	3.05	3.39	2.27	17

**Table 3 molecules-28-00252-t003:** MS/MS parameters for the analysis of target polar compounds in the MRM ESI-negative mode.

Analyte	Type of Pesticide ^a^	Molecular Formula	t_R_ (min)	Quantifier (*m*/*z)*	CE ^b^ (V)	Qualifier (*m*/*z)*	CE (V)	Fragmentor (V)
AMPA	HB	CH_6_NO_3_P	3.04	110 → 63	21	110 → 79	35	116
N-acetyl-AMPA	HB	C_3_H_8_NO_4_P	6.81	152 → 110	10	152 → 63	35	94
Chlorate	HB	ClNaO_3_	5.75	85 → 69	21	83 → 67	21	74
Ethephon	PG	C_2_H_6_ClO_3_P	7.87	143 → 107	10	143 → 79	10	72
HEPA	PG	C_2_H_7_O_4_P	6.20	125 → 95	14	125 → 79	28	98
Fosetyl-Al	FU	C_6_H_18_AlO_9_P_3_	3.24	109 → 81	12	109 → 63	34	90
Glyphosate	HB	C_3_H_8_NO_5_P	8.96	168 → 150	8	168 → 124	10	96
Glufosinate	HB	C_5_H_15_N_2_O_4_P	3.18	180 → 136	16	180 → 63	48	108
N-acetyl-glufosinate	HB	C_7_H_14_NO_5_P	7.86	222 → 136	23	222 → 59	13	116
Maleic hydrazide	PG	C_4_H_4_N_2_O_2_	3.64	111 → 83	12	111 → 82	18	114
MPPA	HB	C_4_H_9_O_4_P	8.40	151 → 133	12	151 → 107	14	104
Phosphonic acid	FU	H_3_PO_3_	11.68	81 → 79	15	81 → 63	35	54
Ethephon D4 (ILIS)		C_2_H_2_ClO_3_PD_4_	7.87	147 → 111	4			60
Fosetyl-Al D15 (ILIS)		_3_C_2_D_5_HO_3_P.Al	3.20	114 → 82	14			66
Glyhosate-^13^C_2_, ^15^N (ILIS)		C^13^C_2_H_8_^15^NO_5_P	17.9	171 → 63	33			102
^18^O_3_-Phosphonic acid (ILIS)		H_3_P^18^O_3_	7.88	87 → 85	19			60

^a^ FU: Fungicide; PG: Plant growth regulator; HB: Herbicide. ^b^ CE: Collision energy.

## Data Availability

Not applicable.

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
