# Peer review of "Monitoring and Exposure Assessment of Fosetyl Aluminium and Other Highly Polar Pesticide Residues in Sweet Cherry"

_molecules, 2022, doi:10.3390/molecules28010252_

Round 1

Reviewer 1 Report

The manuscript titled "Monitoring and exposure assessment of fosetyl aluminium and 3 other highly polar pesticide residues in sweet cherry" By Gormez et al., submitted to Molecules was aiming to determine residues of twelve highly polar pesticides in Turkish cherries intended for export to various countries. 

The conclusion of this study that only phosphonic acid residues, calculated as fosetyl-Al, sum, were detected in cherry samples with no health risk through the consumption of this sweet cherry intended for export from Turkey.

This study is an important for showing the safety consumption of sweet cherry regarding to the most recommended pesticides for its pest control. Accordingly, I recommend publishing of this study after considering the comments indicated the manuscript file attached with this message.

Author Response

Dear Editor,

  • We would lie to thank Reviewer 1 for considering the paper to be worth enough, and publishable after minor revision. The WHOLE manuscript has been revised and the corrections have been made according to reviewers’ suggestions.
  • The Figure 1 has been revised.
  • The Figure 2 has been revised.
  • The term “cyhalothrin, lambda” has been corrected to “lambda-cyhalothrin”.
  • We would lie to thank Reviewer 1 for considering the paper to be worth enough, and publishable after minor revision. The WHOLE manuscript has been revised and the corrections have been made according to reviewers’ suggestions.
  • The Figure 1 has been revised.
  • The Figure 2 has been revised.
  • The term “cyhalothrin, lambda” has been corrected to “lambda-cyhalothrin”.

Reviewer 2 Report

The MS entitled “Monitoring and exposure assessment of fosetyl aluminium and other highly polar pesticide residues in sweet cherry” was reviewed. The authors have summarized their research work on the determination of twelve selective polar pesticides in sweet cherries intended for export. The work is related to food security and noteworthy while the data seems to be original. The experimental procedures are validated and results have been well presented. I suggest some minor corrections/queries.

1. Did the authors investigated the causes of residual Fosetyl-Al? probably if any chemical interaction of this pesticide with cherries is involved, it may be interesting to find out the reasons of accumulation. Add in the discussion section.

2. Did the authors performed some preliminary data collection from the harvesters about the 12 pesticides if those have been in their use? If yes, add in experimental section.

3. Which of the pesticide amongst all does not accumulate in cherries? Again, the data from harvesters is necessary for a fair comparison.

4. Why formic acid was added to methanol in centrifugation? Clarify.

5. A combined LC-MS chromatogram should be provided to compare the same RT with that of pure and control sample. At least one for reader’s interest in the results section.

Author Response

Dear Editor,

  • We would lie to thank Reviewer 2 for considering the paper to be worth enough, and publishable after minor revision. The WHOLE manuscript has been revised and the corrections have been made according to reviewers’ suggestions.

1)Did the authors investigated the causes of residual Fosetyl-Al? probably if any chemical interaction of this pesticide with cherries is involved, it may be interesting to find out the reasons of accumulation. Add in the discussion section.”

Response to Reviewer Comment No 1:The use of fosetyl-Al is not common among cherry farmers in Turkey. Farmers declared that they do not use fosetyl-Al in contrary to foliar fertilizer. Residues of phosphonic acid, defined as fosetyl-Al, in cherries could occur as a result of applying foliar fertilizer containing phosphonic acid itself prior to harvest.

2) Did the authors performed some preliminary data collection from the harvesters about the 12 pesticides if those have been in their use? If yes, add in experimental section.

Response to Reviewer Comment No 2: The authors were not performed preliminary data collection from the harvesters about target analytes. Samples taken from the manufacturers were analysed, and the results were reported.

3) “Which of the pesticide amongst all does not accumulate in cherries? Again, the data from harvesters is necessary for a fair comparison.”

Response to Reviewer Comment No 3:There is no data on pesticides that will specifically accumulate in cherries.

4) “Why formic acid was added to methanol in centrifugation? Clarify.”

Response to Reviewer Comment No 4:Formic acid was used for the adjustment of pH.Extractions with the QuPPe extraction solvent (methanol containing 1% formic acid) were shown to provide quantitative extraction yields of polar substances in cherry.

5) “A combined LC-MS chromatogram should be provided to compare the same RT with that of pure and control sample. At least one for reader’s interest in the results section.”

Response to Reviewer Comment No 5:The figure“LC-MS/MS chromatograms of the extract of the cherry sample containing phosphonic acid (calculated as foestyl-Al) at a level of 1.73 mg kg-1” has been inserted into the text.
